# The Application of Clinical and Molecular Diagnostic Techniques to Identify a Rare Haemoglobin Variant

**DOI:** 10.3390/ijms25126781

**Published:** 2024-06-20

**Authors:** Michela Salvatici, Cecilia Caslini, Simona Alesci, Grazia Arosio, Giuliana Meroni, Ferruccio Ceriotti, Massimiliano Ammirabile, Lorenzo Drago

**Affiliations:** 1UOC Laboratory of Clinical Medicine with Specialized Areas, IRCCS MultiMedica, Via Fantoli 16/15, 20138 Milan, Italy; michela.salvatici@multimedica.it; 2Bianalisi, Clinical Laboratory, Via Mattavelli 3, 20841 Carate Brianza, Italy; cecilia.caslini@bianalisi.it (C.C.); simona.alesci@bianalisi.it (S.A.); grazia.arosio@bianalisi.it (G.A.); giuliana.meroni@bianalisi.it (G.M.); 3Clinical Pathology, Fondazione IRCCS Ca’ Granda Ospedale Maggiore Policlinico, Via F. Sforza 35, 20122 Milan, Italy; ferruccio.ceriotti@policlinico.mi.it (F.C.); massimiliano.ammirabile@policlinico.mi.it (M.A.); 4Clinical Microbiology and Microbiome Laboratory, Department of Biomedical Sciences for Health, University of Milan, Via Mangiagalli 31, 20133 Milan, Italy

**Keywords:** haemoglobinopathy, pulmonary hypertension, cardiovascular risk, laboratory techniques, unusual case, HbA_1c_

## Abstract

Haemoglobin disorders represent a heterogeneous group of inherited conditions that involve at least one genetic abnormality in one or more of the globin chains, resulting in changes in the structure, function, and/or amount of haemoglobin molecules, which are very important for their related clinical aspects. Detecting and characterizing these disorders depends primarily on laboratory methods that employ traditional approaches and, when necessary, newer methodologies essential for solving a number of diagnostic challenges. This review provides an overview of key laboratory techniques in the diagnosis of haemoglobinopathies, focusing on the challenges, advancements, and future directions in this field. Moreover, many haemoglobinopathies are benign and clinically silent, but it is not uncommon to find unexpected variants during routine laboratory tests. The present work reported a rare and clinically interesting case of identification of haemoglobin fractions in an adult man by the determination of glycated haemoglobin (HbA_1c_) during a routine laboratory assessment, highlighting how the correct use of laboratory data can modify and improve the patient’s clinical management.

## 1. Introduction

Haemoglobinopathies are a group of inherited disorders characterized by abnormalities in the structure or production of haemoglobin (Hb), the oxygen-carrying protein in red blood cells. These disorders pose significant diagnostic challenges in laboratory settings due to their complex nature, variable clinical presentations, and the need for accurate identification to guide patient management. In recent years, advancements in diagnostic laboratory techniques have played a crucial role in enhancing the accuracy and efficiency of haemoglobinopathy detection. This review explores key laboratory issues in the diagnosis of haemoglobinopathies, including screening methods, focusing on the challenges, advancements, and future directions in this field.

Screening for haemoglobinopathies typically begins with routine laboratory tests such as complete blood count (CBC), iron metabolism tests, and haemoglobin characterization [1,2,3]. CBC helps to detect anaemia, which is a common manifestation of haemoglobinopathies, while haemoglobin characterization provides information about the types and proportions of different haemoglobin fractions present in the blood. The two most routinely used high-throughput screens for haemoglobin characterization are high-pressure liquid chromatography (HPLC) and capillary zone electrophoresis (CZE) [4,5,6]. Both HPLC and CZE detect haemoglobin species based on their differential elution or migration using spectrophotometric detection at 415 nm. The primary analytical difference between the two methods is their procedure of separation: HPLC separates different haemoglobin variants based on their unique retention times, allowing for quantitative assessment of each variant, while CZE separates proteins based on buffer pH, isoelectric point (pI), and endosmotic flow. These techniques are highly sensitive and can detect both common and rare haemoglobin variants [7,8], but these methods suffer from limitations in detecting rare or complex variants and require confirmatory testing, delaying diagnosis, and treatment initiation [9]. Other laboratory methods include gel electrophoresis and mass spectrometry. The first can help to confirm or further characterize any abnormalities detected using HLPC or CZE [10,11,12]. The second one may improve the sensitivity of Hb analysis as a complementary method to HPLC or CZE or could be used for haemoglobinopathies analysis in large population studies. However, operating and maintaining a mass spectrometer requires a significant degree of expertise. Therefore, further studies will need to be performed to define the best strategy for their use within a clinical laboratory [10,11,12].

While screening tests provide valuable initial information, confirmatory tests are necessary to definitively diagnose specific haemoglobinopathies. Molecular genetic testing, such as polymerase chain reaction (PCR) and DNA sequencing, plays a crucial role in identifying the underlying genetic mutations responsible for haemoglobinopathies. These tests can detect point mutations, deletions, or insertions in the globin genes, providing detailed information about the genotype of the patient [6,13,14,15,16]. During the last few years, advancements in molecular diagnostics, particularly next-generation sequencing (NGS), have revolutionized genetic testing, shifting to a rapid transition from a research setting to a clinical application and becoming the method of choice in many clinical genetics laboratories. NGS-based approaches could enable a comprehensive analysis of haemoglobin gene mutations, including rare and novel variants, not only facilitating precise diagnosis but also providing valuable insights into genotype–phenotype correlations and disease prognosis [17]. Recently, the integration of bioinformatics tools and data analysis algorithms has been demonstrated to be pivotal in interpreting complex genetic data obtained from molecular testing. Bioinformatics platforms enable efficient variant annotation, classification, and interpretation by comparing patient sequences with reference databases and predictive algorithms. Furthermore, machine learning algorithms offer the potential to predict disease outcomes, optimize treatment strategies, and identify novel genetic modifiers. Collaborative efforts to establish standardized databases and algorithms enhance data sharing and facilitate genotype–phenotype correlations, ultimately improving diagnostic accuracy and patient care [18].

Despite notable advancements, several challenges persist in haemoglobinopathy diagnosis, necessitating ongoing research and innovation. One major challenge is the genetic heterogeneity of these disorders, with hundreds of different mutations identified in the globin genes. Laboratories must maintain comprehensive databases of known mutations and constantly update their testing protocols to ensure accurate detection. Access to advanced laboratory technologies remains limited in resource-limited settings, hindering timely diagnosis and management. Additionally, the interpretation of genetic variants requires prompt updates and refinement as new variants are continuously discovered and characterized. Lastly, the integration of multi-omics approaches, including proteomics and metabolomics, holds promise in elucidating disease pathophysiology and identifying novel biomarkers for diagnosis and prognosis [19].

An overview of the main characteristics of the current and emerging approaches for diagnosing and characterizing abnormal haemoglobin variants is reported in Table 1. 

Last but not least, it should be remembered that Hb variants can also be detected accidentally via non-elective methods. Cases of causal findings of haemoglobinopathies during the determination of glycated haemoglobin (HbA_1c_) were reported in the literature [20,21]. HbA_1c_ is haemoglobin that has been irreversibly modified by the addition of glucose via a non-enzymatic process and provides a weighted average of blood glucose concentration over the erythrocyte lifespan. Therefore, it is a biochemical marker routinely used in the management of diabetes mellitus to monitor long-term glycaemic control and assess the risk of developing complications [22,23]. Many commercial methods are available to determine the HbA_1c_ concentration. Those based on electric charge differences (ion exchange HPLC and capillary electrophoresis CE) between HbA_1c_ and other non-glycated forms of haemoglobin represent the methods generally more used for the HbA_1c_ measurements [24]. However, it is known that certain haemoglobin variants can ‘confound’ HbA_1c_ measurements via various mechanisms; it is not uncommon to encounter an anomalous peak or to have falsely elevated or low HbA_1c_ results in chromatograms/electropherograms of patients who undergo this measurement, and then unexpectedly discover to be carriers of a silent variant [25,26,27]. Epidemiological data showed that diabetes prevalence has been rising in recent decades worldwide [28]. Therefore, the unexpected finding of haemoglobin variants in the population that turns to hospitals and laboratories for the measurement of glycated haemoglobin is consequently expected. 

Here, we report a rare and clinically interesting case of a haemoglobin variant in an adult man who underwent a routine laboratory assessment.

## 2. Presentation of an Adult Unusual Case of Haemoglobinopathy

A 58-year-old Caucasian male patient with a previous history of Ankylosing Spondylitis went to a Bianalisi laboratory sampling centre (Carate, Monza Brianza, Italy) to undergo routine biochemical and haematological tests, which included HbA_1c_ and Oral Glucose Tolerance Test (OGTT). Laboratory test results are shown in Table 2. Liver enzymes, haemoglobin, mean cell volume (MCV), mean cell haemoglobin (MCH), and red blood cell (RBC) resulted within reference intervals, while lipids profiles and OGTT showed pathological values. 

The HbA_1c_ determination, performed by the HPLC method (G11st, Tosoh Bioscience, San Francisco, CA, USA) provided the following results: HbA_1c_ wad undetectable due to the presence of a suspected Hb variant (Figure 1). 

Due to the HbA_1c_ result, it was deemed appropriate to investigate the case further by studying the haemoglobin fractions using a specific kit for HbA_2_, HbF, and Hb variants detection. Unexpectedly, the chromatograms did not detect the presence of variants (Figure 2). 

Since the different separation methods available on the market for the evaluation of haemoglobin levels (HPLC vs. electrophoresis) allow for the identification of most of the variants with some differences due to the different chemical–physical principles used, we decided to send our sample to a hospital that employed the capillary electrophoresis method (Capillarys, Sebia, France). The laboratory reported an HbA_1c_ measurement within reference intervals (33 mmol/mol reference, range 20–42 mmol/mol) without indicating the presence of a hypothetical variant (chromatogram not available). 

Therefore, we requested another blood sample from the patient to repeat the analyses both on our HPLC-G11 and on a different HPLC (Variant II—Bio-rad Laboratories, Hercules, CA, USA, XU 2016) in use at Hospital Fondazione IRCCS Ca’ Granda Ospedale Maggiore Policlinico, Milan, Italy. The G11 chromatogram confirmed, as expected, the result obtained on the first sample, that is, it did not allow for the measurement of glycated haemoglobin and suggested the presence of the variant, while the analysis carried out on Variant II provided a glycated haemoglobin result (35 mmol/mol, reference interval 20–42 mmol/mol) and did not show any peaks or anomalies on the chromatogram (Figure 3). 

Analysis of haemoglobin by electrophoresis or HPLC is the most commonly used and widely diffused method to support the diagnosis of haemoglobinopathies. However, occasionally haemoglobin variants can be missed by these methodologies and require additional testing. The persistence of a suspect regarding the presence of a variant, even if the specific diagnostic kit did not detect it, prompted us to undertake a molecular investigation. The patient was informed, and after genetic counselling, he agreed to undergo a second level analysis. He signed the informed consent, and a new blood sample was sent to Oxford University Hospitals for genetic investigations. Testing was undertaken by sequencing the alpha-globin genes (HBA1 e HBA2) and the beta-globin gene (HBB). 

## 3. Molecular Method Diagnosis

The result obtained by molecular diagnosis confirmed that the patient carried abnormal haemoglobin. In particular, molecular analysis of the globin gene highlighted a beta-globin mutation at Codon 5 (CCT>GCT), which gave rise to an anomalous variant of Hb called Hb Görwihl. Haemoglobin Görwihl is a rare haemoglobin variant that was first described in 2003 in the blood sample of a 74-year-old German male having an exceptionally low HbA_1c_ value [29]. Hb Görwihl has functional properties similar to those of normal HbA and, in a heterozygous state, is not associated with clinical symptoms or haematological abnormalities [30]. It only demonstrates altered glycation of the chains with a consequent reduction in the value of glycated haemoglobin; consequently, our patient will not be able to utilize the measurement of HbA_1c_ for his glycometabolic control but will have to measure fructosamine or glycated albumin [31]. 

## 4. Conclusions

The determination of the concentration of HbA_1c_ using HPLC or electrophoretic methods has the advantage of recognizing the presence of abnormal haemoglobins that may have different life spans or, like in the present case, a lower affinity for glucose [29], thus providing misleading information regarding the average glucose concentration. The Tosoh G11 HbA_1c_ kit was the only one able to identify the presence of abnormal haemoglobin, probably due to a higher level of separation of the HbA_1c_ peak that allowed the detection of a peak duplication (see Figure 1), not detected in the other two conditions (Figure 2 and Figure 3). The presence of a haemoglobin variant, when identified, can be compensated and corrected by the software, thus allowing for a correct HbA_1c_ measure, but this event must be checked carefully as it can prevent the release of the result and suggest other tests like fructosamine or glycated albumin.

Therefore, prompt diagnosis, especially in rare cases, can represent a better management of these patients in terms of cardiovascular risk and pulmonary hypertension [32].

In conclusion, the diagnosis of haemoglobinopathies relies on a combination of routine laboratory methods, including complete blood count (CBC), haemoglobin characterization via electrophoresis, high-performance liquid chromatography (HPLC), and molecular, genetic testing. These methods allow the detection, quantification, and identification of haemoglobin variants associated with haemoglobinopathies, facilitating an accurate diagnosis and appropriate patient management. Moreover, it is important to highlight that, in addition to conventional methods for identifying haemoglobinopathies, occasional findings of variants in asymptomatic patients can occur during routine laboratory tests such as the HbA_1c_ measurement. The clinical case described here has demonstrated how the correct management of laboratory data can be crucial to provide a correct clinical assessment of the patient. Therefore, the laboratory should adopt a critical attitude towards the routinely used method, knowing its strengths and limitations as well as evaluating, when appropriate and possible, prompt comparison with a different methodology and the right algorithm. In light of the above, the following is a summary of the algorithm that could be used. In the presence of HBA_1c_ values that cannot be determined or are very low compared to the patient’s clinical conditions, the clinician should ask for further information (especially family history). After a family history examination, or in the absence of additional information, to understand the reason for the low HbA_1c_, molecular analyses should be carried out. In case of silent variants that interfere with HbA_1c_, glycated albumin and/or fructosamine must be necessarily evaluated.

## Figures and Tables

**Figure 1 ijms-25-06781-f001:**
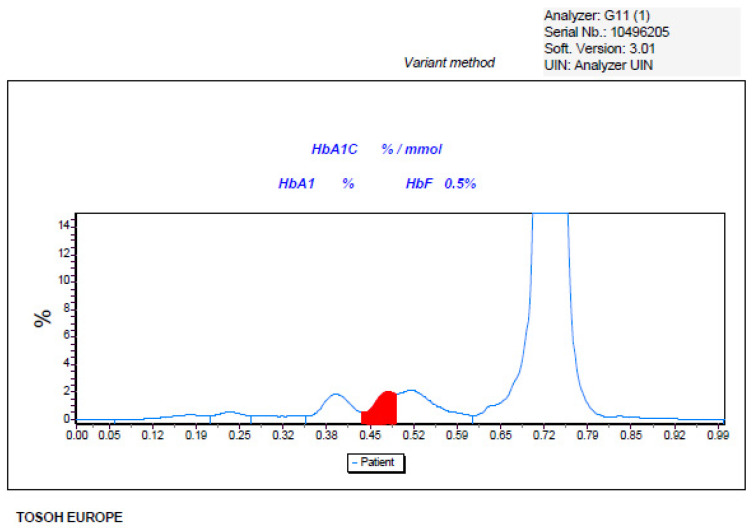
Chromatogram of HbA_1c_ obtained with Tosoh HPLC method.

**Figure 2 ijms-25-06781-f002:**
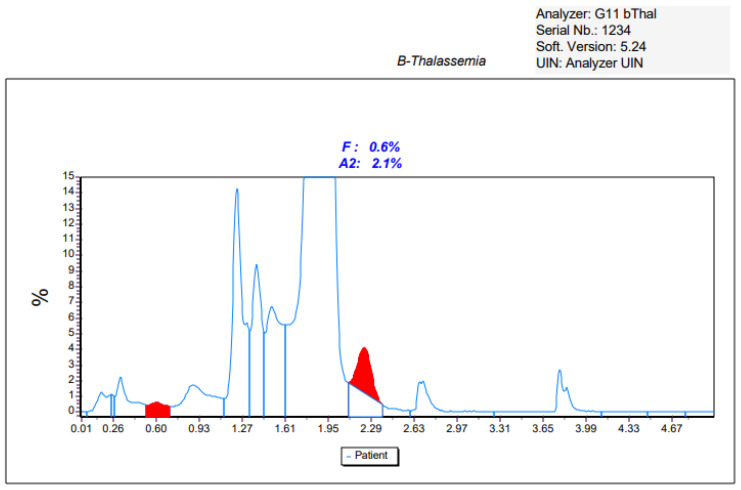
Chromatogram of the studied case, obtained using the Tosoh HPLC method using a haemoglobin kit.

**Figure 3 ijms-25-06781-f003:**
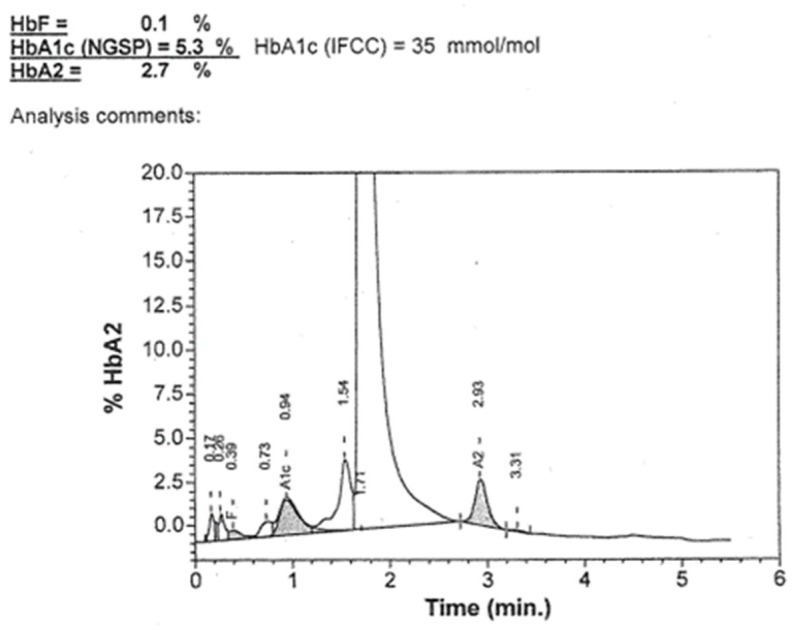
Chromatogram obtained with Biorad HPLC method using HbA_1c_ assay.

**Table 1 ijms-25-06781-t001:** The main characteristics of routine laboratory methods for diagnosing haemoglobinopathies.

Characteristics of Routine Laboratory Methods			
Complete Blood Count (CBC)	High-Performance Liquid Chromatography (HPLC)	Haemoglobin Electrophoresis	Molecular Genetic Testing
This routine test provides valuable information about the cellular components of blood, including red blood cell (RBC) count, haemoglobin concentration, haematocrit, mean corpuscular volume (MCV), mean corpuscular haemoglobin (MCH), and mean corpuscular haemoglobin concentration (MCHC). Anomalies in these parameters, such as microcytosis, hypochromia, or anisocytosis, may indicate the presence of haemoglobinopathies.	This provides high sensitivity, accuracy, and ability to quantify different haemoglobin variants. HPLC separates haemoglobin variants based on their interaction with a chromatographic column and detection system. This method allows for the identification and quantification of both common and rare haemoglobin variants, making it a valuable tool for diagnosing haemoglobinopathies.	This technique relies on the principle that haemoglobin variants migrate at different rates under an electric field due to variations in their charge and size. Haemoglobin variants commonly assessed include haemoglobin A (HbA), haemoglobin A_2_ (HbA_2_), haemoglobin F (HbF), and various abnormal haemoglobins associated with haemoglobinopathies.	Molecular genetic testing plays a crucial role in confirming the diagnosis of haemoglobinopathies and identifying specific genetic mutations responsible for the disorder. Techniques such as polymerase chain reaction (PCR), DNA sequencing, and gene-specific mutation analysis are used to detect point mutations, deletions, or insertions in the globin genes. Molecular genetic testing provides valuable information about the genotype of the patient, which is essential for genetic counselling and family planning.

**Table 2 ijms-25-06781-t002:** Laboratory results.

Parameter	Results	Reference Intervals
Haematology results		
RBC (×10^12^/L)	4.4	4.5–6.00
Haemoglobin (g/L)	146	140–180
MCV (fL)	93	80–95
MCH (pg)	32	26–32
MCHC (g/L)	357	320–360
Biochemistry results		
ALT	24	≤55
AST	19	≤34
Total Cholesterol	215 *	≤190
Triglycerides	185 *	≤150
HDL-cholesterol	40	≥40
OGTT		
Glucose (0 h) mg/dL	112 *	70–100
Glucose (1 h) mg/dL	211 *	<155
Glucose (2 h) mg/dL	108	<140

RBC—Red cell count, MCV—mean corpuscular volume, MCH—mean corpuscular haemoglobin, MCHC—mean corpuscular haemoglobin concentration, AST—aspartate aminotransferase, ALT—alanine aminotransferase, HDL—High-density lipoprotein, OGTT—Oral Glucose Tolerance Test, glucose results before (0 h), 1 hour (1 h) and 2 hours (2 h) after a 75 g oral glucose load. * Out-range results.

## Data Availability

Data are contained within the article.

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
