# Peer review of "The Application of Clinical and Molecular Diagnostic Techniques to Identify a Rare Haemoglobin Variant"

_ijms, 2024, doi:10.3390/ijms25126781_

Round 1

Reviewer 1 Report

Comments and Suggestions for Authors

The authors report an interesting case, along with the necessary diagnostic workout and i have the following concerns:

1. What were the applied techniques for the molecular analysis (lines 172-183, Oxford University Hospital)? Various types of PCRs, Sanger Sequencing, NGS, a combination of these? Please report. 

2. If a patient has the Goerwihl Hb in both alleles (homozygous state for the Goerwihl Hb), what are the expected clinical manifestations? Are there any reported cases in the literature? Please answer that to the discussion. 

3. In your opinion, when there is doubt in an initial electrophoresis, should we proceed with molecular analysis immediately, without losing time for more types of electrophoresis, since i) not all electrophoretic methods will locate the Goerwihl Hb and ii) a final molecular diagnosis for the Goerwihl Hb will be sent for confirming the diagnosis? Please answer that to the discussion. 

4. Abstract, line 24: ....and is not uncommon findings.......can occur......please rephrase the whole sentence because it does not make sense due to grammatical and syntactical errors.  

Author Response

Se cover letter to the Editor.

Reviewer 2 Report

Comments and Suggestions for Authors

The case report entitled  An Unusual Case of Hemoglobinopathy: A Case Report and Review of the Literature provides an overview of key laboratory techniques in the diagnosis of hemoglobinopathies, including complete blood count (CBC), hemoglobin characterization through electrophoresis, high-performance liquid chromatography (HPLC), and molecular genetic testing.

The authors present a rare case of identification of haemoglobin fractions in an adult man by the determination of glycated haemoglobin (HbA1c) during a routine laboratory assessment, highlighting how a correct used of laboratory data can modified and improve the patient clinical management.

Hemoglobin Görwihl is a rare hemoglobin variant with functional properties similar to those of normal Hb A and, in the heterozygous state, is not associated with clinical symptoms or hematological abnormalities. It is associated with altered glycation of the chains with a consequent reduction in the value of glycated hemoglobin, and consequently, the patient will not be able to utilize the measurement of HbA1c for his glycometabolic control but will have to measure fructosamine or glycated albumin.

  I have some minor comments: - What is the practical applicability of the observations raised in the manuscript? - Based on this case report, can the authors come up with one/more algorithms useful in the diagnosis of hemoglobinopathies?

Author Response

See cover letter to the Editor

Reviewer 3 Report

Comments and Suggestions for Authors

This is an interesting case report approached from the perspective of the clinical and molecular diagnostic laboratory.

1. The title should be changed to emphasize that the focus of the manuscript is the challenge in laboratory diagnosis, which would in turn be in keeping with the theme of the special issue/collection. A potential title could be something like "Application of clinical and molecular diagnostic techniques to identify a rare hemoglobin variant". The manuscript is more of an illustrative case report with supportive discussion than it is a literature review.

2. On line 97, the authors refer to diagnosing and characterizing "unstable" hemoglobin variants. "Unstable hemoglobin" has a specific meaning in hematology and it is not clear from looking at Table 1 or from reading the case report, that they are talking about an "unstable hemoglobin" in the hematology sense but rather are talking about an "abnormal hemoglobin" or perhaps an "aberrant hemoglobin".

3. On line 125, the authors refer to Table 1 when they mean Table 2.

4. In Section 3, which globin chain is abnormal? Alpha or beta?

Comments on the Quality of English Language

The English usage issue is entirely about not following capitalization conventions consistently. It would be resolved by a brief professional review.

Author Response

See cover letter to the Editor.
